# Phytochemical, Pharmacological, and Biotechnological Study of *Ageratina pichinchensis*: A Native Species of Mexico

**DOI:** 10.3390/plants10102225

**Published:** 2021-10-19

**Authors:** Mariana Sánchez-Ramos, Silvia Marquina-Bahena, Laura Alvarez, Angélica Román-Guerrero, Antonio Bernabé-Antonio, Francisco Cruz-Sosa

**Affiliations:** 1Departamento de Biotecnología, Universidad Autónoma Metropolitana-Iztapalapa, Av. Ferrocarril de San Rafael Atlixco 186, Col. Leyes de Reforma 1a. Sección, Alcaldía Iztapalapa, Ciudad de México C.P. 09310, Mexico; marianasan_06@hotmail.com (M.S.-R.); arogue@xanum.uam.mx (A.R.-G.); 2Centro de Investigaciones Químicas-IICBA, Universidad Autónoma del Estado de Morelos, Avenida Universidad 1001, Col. Chamilpa, Cuernavaca C.P. 62209, Mexico; smarquina@uaem.mx (S.M.-B.); lalvarez@uaem.mx (L.A.); 3Departamento de Madera, Celulosa y Papel, Centro Universitario de Ciencias Exactas e Ingenierías, Universidad de Guadalajara, Km. 15.5, Carretera Guadalajara-Nogales, Col. Las Agujas, Zapopan C.P. 45020, Mexico

**Keywords:** botanical description, ethnomedicinal uses, standardized extracts, bioactive compounds, biotechnology

## Abstract

*Ageratina pichinchensis* (Asteraceae) has been used for a long time in traditional Mexican medicine for treating different skin conditions and injuries. This review aimed to provide an up-to-date view regarding the traditional uses, chemical composition, and pharmacological properties (in vitro, in vivo, and clinical trials) that have been achieved using crude extracts, fractions, or pure compounds. Moreover, for a critical evaluation of the published literature, key databases (Pubmed, Science Direct, and SciFinder, among others) were systematically searched using keywords to retrieve relevant publications on this plant. Studies that reported on crude extracts, fractions, or isolated pure compounds of *A. pichinchensis* have found a varied range of biological effects, including antibacterial, curative, antiulcer, antifungal, and anti-inflammatory activities. Phytochemical analyses of different parts of *A. pichinchensis* revealed 47 compounds belonging to chromenes, furans, glycosylated flavonoids, terpenoids, and essential oils. Furthermore, biotechnological studies of *A. pichinchensis* such as callus and cell suspension cultures have provided information for future research perspectives to improve the production of valuable bioactive compounds.

## 1. Introduction

For centuries, plants have been an important source of natural products and have a great diversity of uses, including as flavorings, condiments, foods, and medicines [1]. The Asteraceae family is the largest group of angiosperms [2] and is one of the plant families most used in traditional medicine [3,4,5].

Among the members of the Asteraceae, the *Ageratina* genus is the most representative. It can be herbaceous or perennials shrubs and consists of 334 accepted species, including some aggregated from the genus *Eupatorium* [2,6,7]. These species are distributed in tropical and subtropical climates and are characterized by having inflorescences grouped in chapters [5]. The roots and aerial part of *Ageratina glabrata, Ageratina deltoidei*, and *Ageratina adenophora* are used in traditional medicine [8,9,10]. *A. pichinchensis* is especially important because it is a native Mexican plant endemic to the Americas that is a widely used medicinal plant in Mexico [11,12]. Despite its importance, few studies have been conducted to validate only some of its extensive properties. Specifically, they have mainly focused on evaluating its antifungal properties [12,13,14,15,16,17,18,19], its action against different types of ulcers [18,20,21,22,23,24], its wound healing ability [25,26,27], and its antibacterial activity [28]. Recently, through plant biotechnology, extracts of callus and cell suspension cultures were obtained and exhibited anti-inflammatory properties [29,30].

This review aimed to search for studies on the traditional uses, botanical characterization, phytochemistry, pharmacological activity, and biotechnology of *A. pichinchensis*, in addition to providing references and encouraging future research and development regarding this important medicinal plant.

## 2. Research Methodology

Reports on botanical descriptions, traditional uses, bioactive compounds, and pharmacological evaluations of *A. pichinchensis* were collected, analyzed, and summarized in this review. Scientific search engines such as PubMed, ScienceDirect, SpringerLink, Web of Science, Wiley Online, SciFinder, and Google Scholar were used to collect all published articles about this species. The collected manuscripts were identified and examined based on their titles and abstracts. Chemical structures were drawn using ChemDraw Pro 8.0 software. The PubChem database was used to check the IUPAC names of the secondary metabolites reported for this species.

The inclusion criteria for the systematic review of *A. pichinchensis* were the following measures: (1) the article corresponded to a publication in English; (2) the study contained information on the isolation of pure compounds, derivatives, or secondary metabolites; (3) the study contained evaluations of the pharmacological activity of pure compounds, derivatives, or metabolites; and/or (4) the study discussed and reported on traditional or popular uses. The information collected from the revised manuscripts included the name of the authors, the plant species, the part of the plant evaluated, isolated compounds, the extracts and fractions used for biological/pharmacological activity tests, and the experimental technique, concentration, or doses used during biological tests. The data were collected in an Excel sheet and later modified according to the current publication format.

## 3. Botanical Description

It is known that King and Robinson [31] conducted studies of floral anatomy and managed to make a precise delimitation of *Ageratina* from the genus *Eupatorium*. The authors reported that the genera that are most closely related to *Ageratina* are not necessarily those that have been confused with it and they often have distinctive characteristics and have been kept separate from *Eupatorium*.

*A. pinchinchesis* (Figure 1) (formerly *Eupatorium pichinchense*, *Ageratina aschenborniana*, or *Eupatorium aschenbornianum*) is described as a shrub up to 1.5 m tall; its stem is erect, highly branched, cylindrical, yellowish-brown, occasionally a little reddish, glabrous, or slightly puberulent. The opposite leaves have a petiole 1.5 to 2 cm long and are puberulent, lamina ovate, trinervate from the base, and 3 to 6 cm long by 1.5 to 4 cm wide, with a sharp apex, a crenate-toothed edge, a rounded base, a pubescent upper surface and lower underside, prominent veins, and membranous. It has numerous ± 5-mm-long chapters arranged in terminal and axillary compound corymbs, with pubescent pedicels and a campanulate involucre ± 4 mm long by ± 3.5 mm wide that is slightly shorter than the flowers. Its bracts are arranged in three series of similar length, and are oblong-lanceolate, obtuse, green (the external ones are reddish), and pubescent, with two prominent nerves. It bears 20 to 40 flowers with a white corolla ± 3.5 mm long and pubescent on the lobes; the achene is 1.5 to 2 mm long and hairy, and the pappus is the same length as the corolla, with white bristles [7,32].

Here, we must clarify that due to the issues with nomenclature described above, throughout the document, we will refer to studies on *A. pichinchensis*, including references cited as *E. pichinchense*, *A. aschenborniana*, or other synonyms [31,32].

## 4. Geographic Distribution

The Asteraceae family is the largest group of the Angiosperms [2]. The family has the largest number of described and accepted species (about 24,000) and 1600–1700 genera distributed throughout the world, except Antarctica [33]. In Mexico, the *Ageratina* genus is one of the most relevant with a total of 164 species reported, 135 of which are endemic and 41 are micro-endemics; these plants can be found in the temperate forest and the xerophilous scrub [4]. All *Ageratina* species, including *A. pichinchensis*, are native to Mexico, Central America, and South America [34,35]. In fact, *A. pichinchensis* is one of the representative species of the *Ageratina* genus in the Americas (Figure 2A), distributed in the following order of importance: Mexico, Ecuador, Costa Rica, Colombia, Guatemala, Panama, Nicaragua, Peru, Honduras, and Venezuela [36].

*A. pichinchensis* is registered in various states of Mexico (Figure 2B), including Chiapas, Coahuila, Colima, Mexico City, Durango, Guanajuato, Guerrero, Hidalgo, Jalisco, State of Mexico, Michoacán, Morelos, Nayarit, Nuevo León, Oaxaca, Puebla, Querétaro, Sinaloa, Sonora, Tamaulipas, Tlaxcala, and Veracruz [36,37].

## 5. Ethnomedicinal Uses

For centuries, the *Ageratina* genus has been used in traditional medicine in several countries [38,39]. The Native American Ethnobotany database mentions that the Cherokee tribe used this plant against diarrhea, urinary diseases (diuretics), and fever and as a stimulant and tonic. On the other hand, the Iroquois used a decoction of the roots as a sweat bath to keep a sick person cool, and an infusion of the roots to deflate the uterus, whereas the Chickasaw used chewed roots held in the mouth against toothache, and the Choctaw used it as a tonic and stimulant against heat stroke. The Meskwaki used it as a moisturizing agent in steam baths and to awaken an unconscious patient. The Navajo-Ramah used a cold infusion taken as a lotion for headaches and fever, while the Zuni used it to make a paste that was applied externally for rheumatism and swelling [10].

In Mexico, various *Ageratina* species are commonly named *axihuitl*. The Ethnobotanical Garden and Museum of Traditional and Herbal Medicine of the National Institute of Anthropology and History of Mexico (INAH) mention that this plant is reported in 16th century sources, e.g., in the books of the protomedic Francisco Hernández, this species is cited as *axihuitl*, *apatli*, or a water remedy [40]. Various traditional uses are reported in the Digital Library of Traditional Mexican Medicine; an infusion of the root is taken to relieve stomach pain, the cooked aerial parts are used postpartum, as a purgative, and to treat swelling, liver diseases, dysentery, gastritis, indigestion, and colic in the kidneys [11]. On the other hand, in regions of Nepal, it was reported that *A. adenophora* leaves are used to make a paste that is used to massage the crooked parts to relieve pain and swelling, while the juice of the leaves is applied to cuts and wounds for its hemostatic and antiseptic properties [10,41]. It has also been mentioned that the aerial parts of *A. pichinchensis* are popularly used to treat skin problems, wounds, tumors, and canker sores [13]. In fact, according to interviews with empirical midwives and some herbalists in rural areas of the State of Morelos, Mexico, this plant is used against respiratory, genital, gastrointestinal, urinary, and skin infections [12].

## 6. Phytochemistry

Medicinal plants are the principal reserve for a variety of bioactive molecules which can be a source for discovering and developing new therapeutic agents. The pharmacological potential of a medicinal plant depends on its secondary metabolites. The secondary metabolites of *A. pichinchensis* have been the subject of several studies, and all these studies have focused on the aerial biomass. Phytochemical screening of its extracts and fractions has demonstrated a wide variety of phenolic compounds. Among the main compounds reported for *A. pichinchensis* are benzochromenes, benzofurans, glycosylated flavonoids, and terpenes (Table 1). On the other hand, the essential oil from *A. pichinchensis* leaves has been analyzed by gas chromatography-mass spectrometry (GC-MS), in which 25 compounds (Table 1) were identified by comparing their mass spectra with the GC-MS library data and by their retention indices (RI).

### 6.1. Benzochromenes and Benzofurans

These groups of compounds are generally present in the leaves or stems (up to 5% of the weight) and they are less frequent in the roots. These compounds are biogenetically formed by combining an isoprene unit and a phenolic system. The formation of the heterocyclic ring can occur in two different ways, yielding 2,2-dimethylchromene or 2-isopropylbenzofuran. Most of them have a methyl ketone group, usually in the para position to the oxygen function of the heterocyclic ring. Until recently, benzofurans comprised a very small group of natural products and the first reported member of this group was the euparin (6-hydroxydehydrotremetone) compound isolated from *Eupatorium purpurem* [45]. Many chromenes and benzofurans have been shown to be biologically active; for example, toxol and dehydrotremetone are bacteriostatic; tremetone, dehydrotremetone, and hydroxytremethone are toxic to goldfish; and the toxol and toxyl angelate exhibit weak antitumor activity against P-388 lymphocytic leukemia [46].

Chromenes (benzopyrans) represent an important class of secondary metabolites because of their interesting biological and pharmacological properties [47]. For instance, several studies have demonstrated the antibacterial, anti-inflammatory, antioxidant, and cytotoxic activities of these compounds [48,49,50,51]. Furthermore, some of these compounds have been reported to act against Alzheimer’s disease [52,53]. The content and type of the main compounds vary considerably between samples, depending on the origin of the plants. Even the time of collection is a factor that modifies the chemical composition of the extracts.

Several chromenes and benzofurans are volatile and can be extracted by steam distillation; however, extracting fresh or dried plants with organic solvents, such as benzene, petroleum ether, ethyl ether, dichloromethane, chloroform, or ethyl acetate, is a more suitable procedure.

The first phytochemical study for *A. pichinchensis* was carried out by Gómez et al. [42]. In this work, the compounds named eupatoriochromene B (**1**) and eupatoriochromene C (**2**) were isolated and identified for the first time. In addition, a new benzofuran derivative (**8a**) was isolated. They also obtained the known compounds 7-methoxy-2,2-dimethylchromeno (**3**), 5-hydroxy-6-acetyl-8-methoxy-2,2-dimethylchrome (**4**), 5,8-dimethoxy-6-acetyl-2,2-dimethylchromene (**5**), and 2-isopropenyl 6-methoxy-2,3-dihydrobenzofuran (**7**). Compounds **6**, **8b**, **9**, and **10** were obtained as derivatives of Compounds **1**, **2**, and **8a**; these allowed researchers to establish its chemical structure unequivocally. In order to establish the chemical composition of *n*-hexane extract, Aguilar-Guadarrama et al. [16] used successive chromatographic procedures and spectroscopic and spectrometric data to isolate and identify two new benzofurans, encecalinol (**11**) and encecalin (**12**), as the main products, and 5-acetyl-3*β*-angeloyloxy-2*β*-(1-hydroxyisopropyl)-2,3-dihydrobenzofurane (**13**) and 5-acetyl-3*β*-angeloyloxy-2*β*-(1-hydroxyisopropyl)-6-methoxy-2,3-dihydrobenzofurane (**14**), *O*-methylencecalinol (**15**), and sonorol (**16**) as the minor compounds. They standardized the extracts by HPLC analysis to determine the content of the compounds.

Sánchez-Mendoza et al. [20] obtained three organic extracts of the dried leaves (hexane, dichloromethane, and methanol) and evaluated the extracts in an experimental model of gastric ulcers induced by ethanol in rats. They showed that the hexanic extract was the most gastroprotective. By column chromatography purification of the extract, encecanescin (**17**) was obtained, proving to be the main active gastroprotective compound. A recent study conducted by Torres-Barajas et al. [44] reported the 6-acetyl-7-hydroxy-8-methoxy-2,2-dimethylchrome compound (**18**) and two of its derivatives (**19** and **20**) (Figure 3 and Table 1).

On the other hand, Sánchez-Mendoza et al. [21] and Aguilar-Guadarrama et al. [16] demonstrated the presence of other secondary metabolites, such as acetophenones, terpenes, and sterols, in *A. pichinchensis* (Figure 4 and Table 1).

### 6.2. Phenolic Compounds

Phenolic compounds are secondary metabolites present in several plants that have important biological activities. These compounds can be classified as flavonoids, phenolic acids, and other conjugate compounds [54]. Flavonoids play an important role in nature, mainly due to the chelating potential conferred by their chemical structure. In humans, this compound has a potent antioxidant effect, which is associated with reducing the risk of certain chronic diseases, preventing some cardiovascular diseases and some types of cancer [55,56].

In some Asteraceae species, such as *Arnica montana* and *Arnica chamissonis*, important phenolic compounds have been found, e.g., chlorogenic acid (**29**), 7-*O*-(*β*-D-glucopyranosyl)-gossypetin (**30**), and 7-*O*-(*β*-D-glucopyranosyl)-galactin (**31**) (Figure 4 and Table 1) [25,26]. Traditionally, these plant species have been used to treat wounds [56]. *A. pichinchensis* it is a closely related species, of which the compounds **29**, **30**, and **31** were also found.

### 6.3. Chemical Constituents of the Essential Oil

In one of the first reports on the composition of the essential oils of *A. pichinchensis* leaves obtained by the hydrodistillation method, a GC/MS analysis revealed that the oil is mainly composed of 8,9-epoxythymyl isobutyrate (**40**, 20.2%), germacrene-D (**37**, 19.8%), thymyl isobutyrate (**38**, 10.8%), eupatoriochromene (**1**, 6.5%), and encecalol (**44**, 5.9%) compounds (Table 1, Figure 5). There is only one report on the composition of essential oils of *A. pichinchensis* [28].

## 7. Pharmacological Activities

*A. pichinchensis* has different pharmacological activities, including antimicrobial, anti-inflammatory, antiulcer, and healing properties. Biological studies have corroborated its use in traditional medicine to treat various ailments. A few studies have also reported its activities related to wound healing. Some of these activities are explained in detail in the following subsections.

### 7.1. Antimicrobial Activity

#### 7.1.1. In Vitro Assays

In a first study, hexane and methanolic extractions were carried out with the aerial parts of *Asclepia curassavica*, *Bixa orellana*, *E. aschenbornianum* (currently *A. pichinchensis*), *Galphimia glauca*, *Lysiloma acapulcensis*, *Malva parviflora*, *Sedum oxypetalum*, and *Senecio angulifolius* and the extracts were evaluated against microorganisms. In this study, it was found that the hexanic extract of *A. pichinchensis* stood out for being more effective against *Trychophyton mentagrophytes* (MIC of 0.03 mg/mL); for *Candida albicans* and *Aspegillus niger*, the MIC values were 8.0 and 4.0 mg/mL, respectively [12]. These results led to another study on the isolation and chemical characterization of the chemical constituents of hexane extracts of *A. pichinchensis*, of which the compounds **11**, **13, 14**, **22**, **27,** and **28** were identified; an antimicrobial evaluation showed a MIC of 200 µg/mL for Compound **13** against *T. mentagrophytes* and 50 µg/mL against *Trichophyton rubrum*, while Compound **14** revealed a MIC of 50 µg/mL against both dermatophytes. It should be noted that in this work, Compounds **1** and **2** were reported for the first time. Additionally, Compound **22** presented a MIC of 100 µg/mL; however, Compound **11** was outstandingly active against all microorganisms tested, with MIC values of 12.5 µg/mL for *T. mentagrophytes* and *T. rubrum*, 100 µg/mL for *C. albicans*, and 200 µg/mL for *A. niger* [13].

Aguilar-Guadarrama et al. [16] reported the antifungal effect of compounds isolated from the hexane extract of aerial parts of *A. pichinchensis* against dermatophytes that cause athlete’s foot (tinea pedis). Compound **12** had a MIC of 6.2 µg/mL against *T. rubrum* and 12.5 µg/mL against *T. mentagrophytes*, while Compounds **26** and **23** presented a MIC of 12.5 and 25 µg/mL for both microorganisms, respectively. In previous studies, it was shown that Compound **11** had a MIC of 12.5 µg/mL for both *T. rubrum* and *T. mentagrophytes* [13]. Torres-Barajas et al. [28] also reported the antimicrobial effect of a hydroalcoholic extract of *A. pichinchensis* leaves, and by gas chromatography coupled with mass spectrometry, they identified Compounds **40**, **37**, **38**, **41**, and **44** as the main compounds. The extract showed slight inhibition (MIC of 104 mg/mL) against *Staphylococcus aureus* and *Enterococcus faecalis*. In a second study, Compound **11** was isolated from a hexane extract of the aerial parts, which has been reported to be active against *T. macrophytes* and *T. rubrum* [13,14], and thus, could make an important contribution to the search for bioactive compounds against mycosis [44].

#### 7.1.2. Clinical Trials

Romero-Cerecero et al. [14] developed a standardized pharmaceutical formulation containing 10% of the depigmented hexanic extract of *A. pichinchensis* leaves, and its efficacy and tolerability were determined in a double-blind pilot clinical study in patients diagnosed clinically and mycologically with *T. rubrum* and *T. mentagrophytes.* One patient group was treated with the extract and another patient group was treated with ketoconazole (2%), and both were monitored weekly for 4 weeks. The authors found no significant differences in efficiency and tolerability between patients using the extract or the control. The effect shown by the pharmaceutical formulation was reported to be attributed to the encecalinol compound.

In a later study, this same research group developed a standardized 10% lacquer with a depigmented extract of the aerial parts of *A. pichinchensis* and conducted a double-blind randomized clinical study of patients diagnosed with cutaneous mycosis with less than 50% fungal infection, using ciclopirox (8%) as a positive control. The lacquer was applied every 3 days during the first month; in the second month, it was applied twice a week; and from the third month, it was applied once a week. The group treated with the standardized extract showed a therapeutic and mycological efficacy of 71.1 and 59.1%, respectively, while the group treated with the control had 80.9% and 63.8%, respectively, without observing side effects. The authors concluded that the pharmacological effect could be attributed to Compound **12,** as it was the major compound [15].

In another study, by preparing a lacquer (previously decolorized) in two formulations (12.6% and 16.8% extracts), Romero-Cerecero et al. [17] evaluated these treatments against *T. rubrum* (MIC 125 µg/mL) and *T. mentagrophytes* (MIC 250 µg/mL). In addition, these treatments were evaluated for 6 months in patients diagnosed with mild and moderate onychomycosis with 1 to 10 infected nails. The patients treated with lacquer (16.8% extract) showed a statistically higher recovery than was observed with lacquer containing 12.6% extract, and no fungi were found in the mycological microscopic analyses.

On the other hand, in a clinical study with patients with signs and symptoms of erythema, scaling, fissures, itching, erosions, and bad odor, the therapeutic efficacy and tolerability of a pharmaceutical presentation combined with an extract of the aerial parts of *A. pichinchensis* obtained with hexane and ethyl acetate (7:3) (HEA-7:3) containing 0.76% or 1.52% of Compound **12** was tested, with ketoconazole (2%) as a control. The results showed that the clinical efficacy of the treatments and the control were 39%, 48.8%, and 46.51%, respectively, while the mycological efficacy was 68.29%, 72.09%, and 76.74%, respectively. The therapeutic cure was 34.14%, 41.8%, and 39.53%, and for all cases, the tolerability was 100%. Although there were no significant differences among the treatments, the authors mention that the formulation with the highest Compound **12** content was the most favorable [18].

In another case, a clinical study was carried out on vaginal suppositories made with 7% HEA-7:3 containing approximately 15 mg of Compound **12** in female patients diagnosed with vulvovaginal candidiasis. Application of suppositories for 6 days revealed a therapeutic efficacy of 81.2%, while the control of clotrimazole (100 mg) was 86.6%; the tolerability was 94.1% for patients treated with the extract and 93.7% for those treated with clotrimazole [19].

The pharmaceutical formulation of a lacquer based on HEA-7:3 standardized with 24.14 mg/mL of Compound **12** was used and administered for 6 months to patients diagnosed with diabetes mellitus and onychomycosis with 1 to 6 nails infected by *T. rubrum*, *T. mentagrophytes*, *Epidermophyton floccosum*, and *C. albicans*. In all cases, the severity of the condition decreased, i.e., there was a decrease in the number of infected nails (from 0 to 2) and there were no statistical differences with respect to the control (8% ciclopirox lacquer). The authors demonstrated that glucose level control was decisive in the effectiveness of the treatment, so that the clinical effectiveness of the group treated with the extract was 78.5%, compared with 77.2% for the group treated with ciclopirox [57].

### 7.2. In Vivo Antiulcer Assay

Sánchez-Mendoza et al. [20] reported a bio-directed study of the antiulcer effects in rats provoked with absolute ethanol, using hexane, dichloromethane, and methanol extracts from the aerial parts of *A. pichinchensis* (*E. aschenbornianum*). All treatments were active; however, the hexane extract (85.65 ± 4.76%) showed an outstanding activity at 100 mg kg^−1^ with a dose–response effect compared with the other treatments, while the positive control had an effect of 72.88 ± 5.85%. On the other hand, one of the fractions of the hexane extract showed a 68.62 ± 7.81% gastroprotective effect, and the phytochemical study showed that it contained Compound **17** as the major compound.

In a second study, Sánchez-Mendoza et al. [21] also reported the antinociceptive and gastroprotective activity of Compound **21** isolated from leaves via hexane extraction in models of inflammatory and neuropathic pain, and a reduction of nociception was obtained with an efficacy of 72.6% and 57.1% at doses of 100 and 562 mg/kg, respectively. Additionally, there was a maximum gastroprotective inhibition of 75.59% at 100 mg/kg. In both studies, it was shown that the mechanism of action was correlated with endogenous nitric oxide, prostaglandins, and sulfhydryl groups.

Another study reported the antiulcer effects, lipid peroxidation properties, and acute toxicity of dried stems of *A. pichinchensis*. The results indicated that administration of *A. pichinchensis* exerted an antiulcerative effect and decreased lipid peroxidation in gastric ulcers induced with acetylsalicylic acid. The acute toxicity test indicated normal behavior without significant variation in the weight and food consumption of the animals; additionally, a quantitative analysis of the biochemical parameters did not reveal liver or kidney damage, which suggests that the extract may be a safe therapeutic agent for the prevention of gastric ulcers [58].

### 7.3. Wound Healing Activity

#### 7.3.1. In Vivo Assays

The wound healing capabilities of medicinal plant extracts can be beneficial, as pathogenic infections can be prevented if there is early wound healing [59]. Romero-Cerecero et al. [25] reported the wound healing potential of *A. pichinchensis* leaf extract in Sprague–Dawley-strain female rats weighing an average of 250 g. The incision wound healing model was tested. Different groups of animals were treated daily for 8 days with local application of an aqueous extract of leaves from *A. pichinchensis*. Dried extracts were mixed directly in a previously hydrated gel until a 10% concentration was obtained. A positive control group was used; these animals were administered an ointment containing fibrinolysin 1 U (Loomis)/g via the same route and for the same duration. The effect produced by the treatment was higher than exhibited by the positive control. It is noteworthy that in the case of treatment with the aqueous *A. pichinchensis* extract, at the end of the period of administration, the wounds had completely closed in all cases. Based on these results, a bio-guided purification revealed that 7-*O*-(*β*-D-glucopyranosyl)-galactin (**31**) was the major compound associated with the effects of *A. pichinchensis* in cell proliferation [26]. Later, two extracts (aqueous and hexane) with standardized concentrations of 7-*O*-(*β*-D-glucopyranosyl)-galactin were shown to promote the healing of skin lesions in rats with streptozotocin-induced diabetes [27].

#### 7.3.2. Clinical Trials

The healing properties of this plant were also assessed in human clinical trials. The effectiveness of a standardized extract of *A. pichinchensis* was proven to heal chronic venous leg ulcers [18]. In another study, a cream containing an extract of *A. pichinchensis* was topically used by diabetic patients with foot ulcers; the results showed that this treatment decreased the healing time and lesion size, although no significant differences were observed [22].

## 8. Plant Biotechnological Studies on *A. pichinchensis*

The secondary metabolites play an important role in plants, including as a defense mechanism [60]; moreover, they represent a source of bioactive molecules [61]. However, obtaining bioactive secondary metabolites by traditional culture methods may be economically unfeasible [62]. Therefore, it is necessary to produce bioactive compounds of commercial and/or medicinal importance, independent of climatic conditions, the region of production, or other factors. In this regard, plant tissue cultures, e.g., callus or cell suspension cultures, are an alternative for producing secondary metabolites of interest; in addition, it is an advantageous system compared with chemical extraction of the whole plant [63,64].

Although it has been reported that in vitro cultures of several plants produce bioactive compounds, many wild plants continue to be a direct source for obtaining secondary metabolites, for which it is necessary to continue establishing sustainable systems. There are reports that *A. pichinchensis* has healing, antifungal, and antigastritis properties; in addition, it produces bioactive compounds, such as benzochromenes, benzofurans, sterols, and glycosylated flavonoids. In fact, there is a patent that supports its popular use [13,16,26]. However, despite its medicinal importance, this species is the only one of the genus *Ageratina* that has had two biotechnological studies using cell cultures. According to our experience with *A. pichinchensis*, a general scheme for the establishment of a callus and cell suspension culture for this species is presented in Figure 6.

### 8.1. Callus Culture

In the first biotechnological study for *A. pichinchensis*, Sánchez-Ramos et al. [29] developed friable callus cultures. In this study, it was found that extracts with ethyl acetate reduced edema in mice by 35.11 ± 3.79% at 0.1 mg/ear. From these extracts, the compounds (+)-*β*-eudesmol acetate (**48**), demethoxyencecalin (**49**), (*2S*,*3R*)-5-acetyl-7,3α-dihydroxy-2*β*-(1-isoprenyl)-2,3-dihydrobenzofuran (**50**), *β*-sitosterol (**51**), stigmasterol (**52**), *β*-amyrin (**53**), 3-epilupeol (**54**), stigmasterol *β*-D-glucopyranoside (**55**), campesterol (**56**), *n*-hexadecanoic acid (**57**), and hexadecenoic acid methyl ester (**58**) were isolated (Figure 7). In addition, Compounds **50** and **54** showed a significant anti-inflammatory effect [29].

### 8.2. Cell Suspension Culture

In a second biotechnological study, Sánchez-Ramos et al. [30] established a suspension cell culture of *A. pichinchensis*. Compounds **50** and **54** were identified by GC-MS, which were produced in less time than in a callus culture. The extracts showed significant inhibition of nitric oxide (NO). In addition, Compounds **49**, **51**, **52**, **57**, **58**, (-)-artemesinol (**59**), (-)-artemesinol glucoside (**60**), 24-methylene-9,19-cyclolanasthan-3*β*-ol (**61**), 24-methylenecyloartan-3-one (**62**), encecalin (**12**), 3-5-diprenyl-acetophenone (**21**), isopropyl palmitate (**63**), palmitamide (**64**), oleamide (**65**), and methyl pyroglutamate (**66**) were identified (Figure 7).

## 9. Discussion

Fungi and bacteria are found naturally in the human body, especially in the mouth, vagina, intestines, lungs, skin, hair, and nails, affecting health throughout the world [65]. Although there are antimicrobial drugs such as allylamines, azoles, benzoxaborols, cyclo-pyrox, and amorolfine, these can cause adverse effects and microbial resistance, e.g., *Candida* is resistant to azoles [66,67].

On the other hand, the withdrawal of the drug recognized as oral ketoconazole for the treatment of superficial infections was reported by the regulatory authorities in Europe and the United States, since, on some occasions, risks of toxicity have been found [67]. Furthermore, all these infections lead to an inflammatory response, leading to serious abnormalities in body systems. These disadvantages have been overcome by using plant-derived drugs or therapies as complementary and alternative medicine [66]. Medicinal plants and/or their derivatives are considered beneficial due to their properties, that is, their satisfactory effect, ease of availability, low cost, fewer or no side effects, safety, and higher efficiency compared to their synthetic counterparts [68].

In this regard, *A. pichinchensis* is known to have been used extensively in traditional medicine [1,38,39]. Despite its wide distribution, few studies have been carried out to validate its uses in traditional Mexican medicine. Although there have been some phytochemical and pharmacological studies, several of them depended on the aerial parts of the wild plant as the main source of extracts or compounds. Several compounds have been isolated from this species, but others have been obtained by acetylation, hydrogenation, ozonolysis, iodination, or epoxidation of some of these compounds [16,42,44]. The biologically most prominent compounds are encecalinol (**11**), encecalin (**12**), encecanescin (**17**), 3,5-diprenyl-4-hydroxyacetophenone (**21**), 7-*O*-(*β*-D-glucopyranosyl)-gossypetin (**30**), and 7-*O*-(*β*-D-glucopyranosyl)-galactin (**31**), which have mainly antifungal, antiulcer, wound healing, and some antibacterial and anti-inflammatory activities [12,13,14,15,16,17,18,19,20,21,22,23,24,25,26,27,28].

Based on these findings, we believe that *A. pichinchesis* has medicinal potential and further phytochemical and pharmacological studies are necessary, since to date, these have focused on antimicrobial studies, specifically antifungal studies. However, once the extracts or compounds have been characterized, identified, and isolated, and the biological effects have been determined, one of the problems that can arise is the lack of plant material, since plant production depends on factors such as the seasons of the year [63,69], which may limit future studies. In this regard, in this review, we included the only existing studies in the literature on cell cultures, which do not depend on external climatic factors; however, more studies on larger-scale in vitro production are required [70].

## 10. Conclusions

There are few phytochemical studies on *A. pichinchensis*, and these have corroborated its traditional use through clinical and in vitro studies; however, in all of them, the wild plant has been used as a source of compounds or extracts, which is not ecologically viable. On the other hand, there are only two reports of anti-inflammatory activity using extracts obtained from biotechnological cultures, which indicates the need to continue strengthening the study of this species to validate other biological activities of interest, for example, its anti-inflammatory, antibacterial, antioxidant, cytotoxic, antimalarial, or anti-dengue effects. Furthermore, it is important to continue exploring and exploiting the technique of plant cell culture so that, in the future, it will be possible to offer sustainable bioactive products.

## Figures and Tables

**Figure 1 plants-10-02225-f001:**
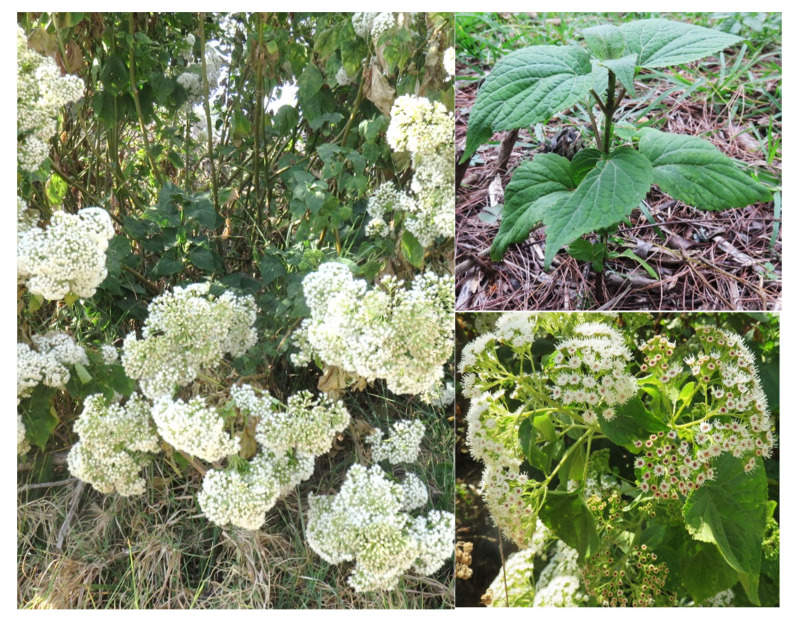
Typical wild plants of *A. pichinchensis* growing in the municipality of Tepoztlán in the state of Morelos, Mexico.

**Figure 2 plants-10-02225-f002:**
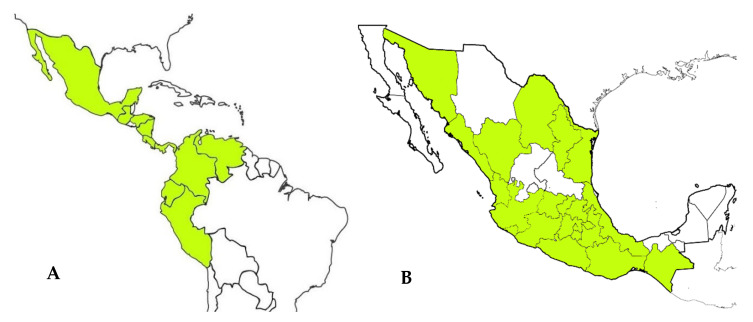
Geographical distribution of *A. pichinchensis* in Latin America (**A**) and Mexico (**B**).

**Figure 3 plants-10-02225-f003:**
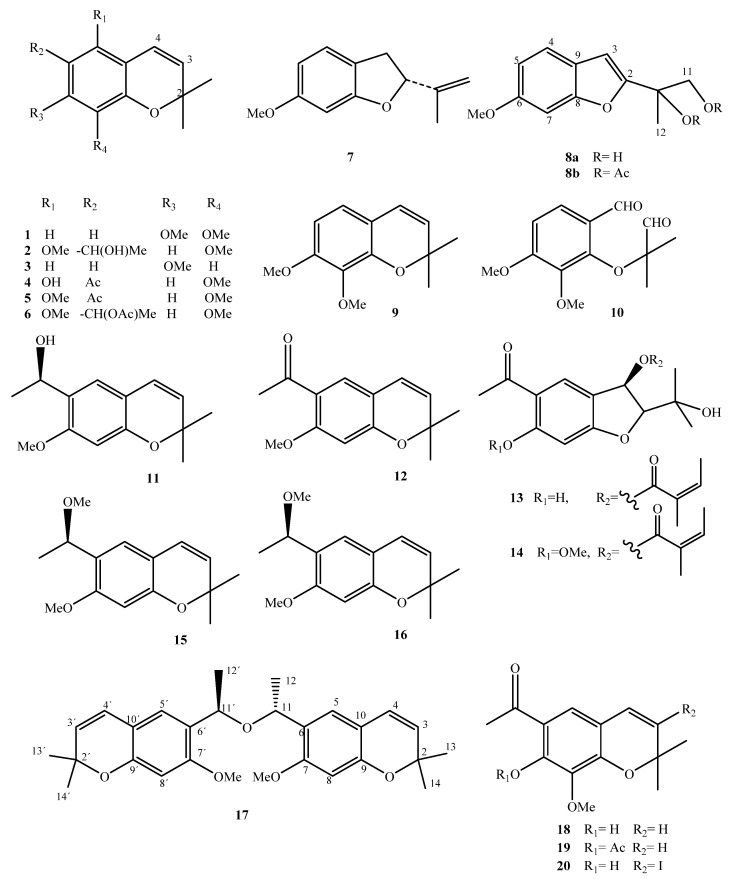
Chemical structure of benzochromenes and benzofurans isolated from *A. pichinchensis*.

**Figure 4 plants-10-02225-f004:**
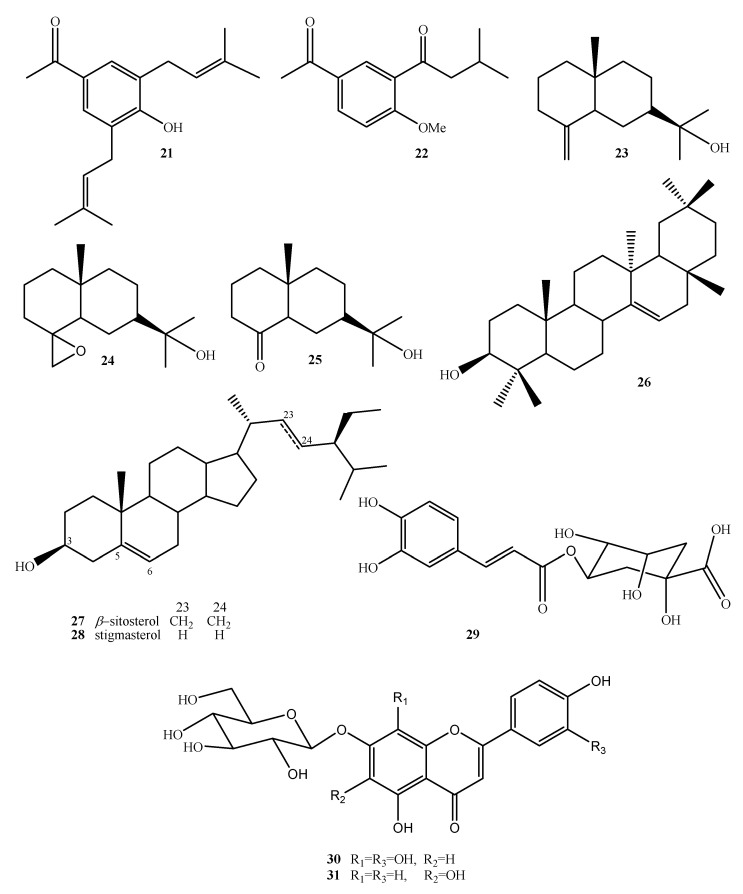
Chemical structure of terpenes and phenolic compounds isolated from *A. pichinchensis*.

**Figure 5 plants-10-02225-f005:**
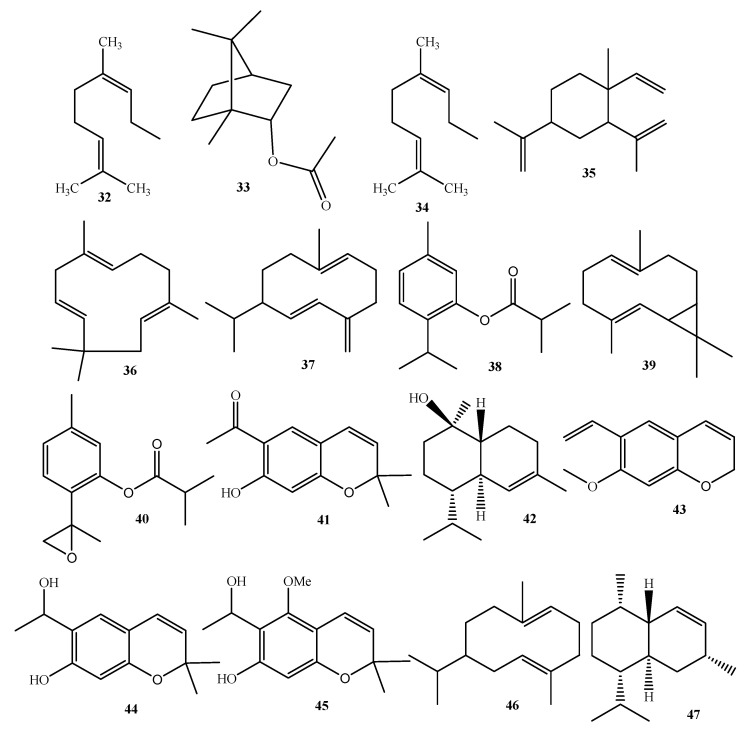
Structures of the reported chemical compounds of essential oils from *A. pichinchensis*.

**Figure 6 plants-10-02225-f006:**
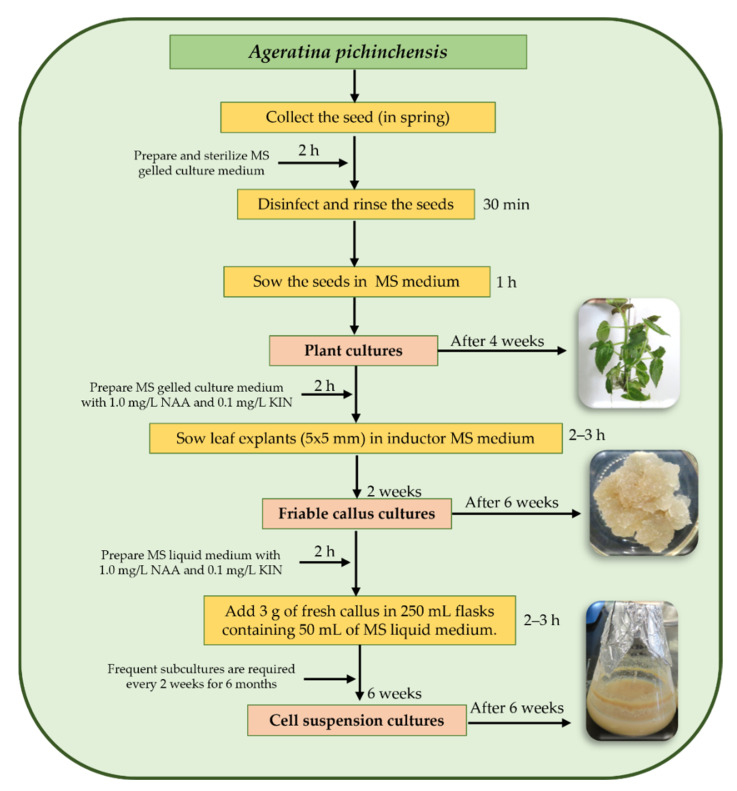
Representation of a general scheme for establishing a callus and cell suspension culture for *A. pichinchensis*. NAA: α-naphthaleneacetic acid; KIN: 6-furfurylaminopurine.

**Figure 7 plants-10-02225-f007:**
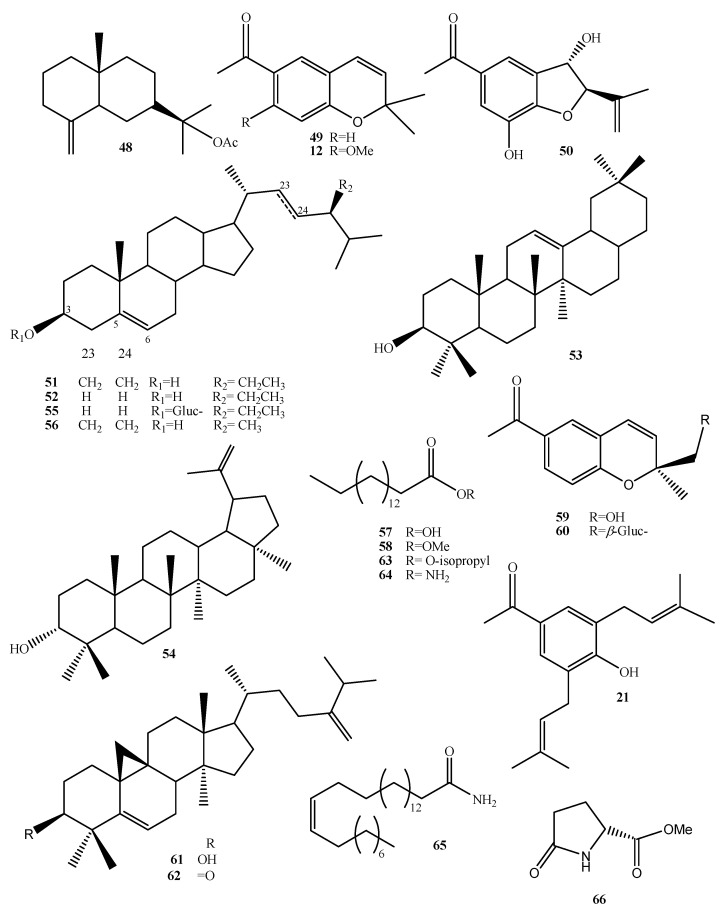
Compounds identified in callus and cell suspension cultures of *A. pichinchensis*.

**Table 1 plants-10-02225-t001:** Chemical constituents isolated from different parts of *A. pichinchensis*.

Compound Name	No.	Plant Source	Extraction Method	Refs.
7,8-Dimethoxy-2,2-dimethylchromene	1	Leaves and flowers	Isolation/CC	[42]
6-[1-Hydroxyethyl]-5,8-dimethoxy-2,2-dimethylchromene	2	Leaves and flowers	Isolation/CC	[42]
7-Methoxy-2,2-dimethylchromene	3	Leaves and flowers	Isolation/CC	[42]
5-Hydroxy-6-acetyl-8-methoxy-2,2-dimethylchromene	4	Leaves and flowers	Isolation/CC	[42]
5,8-Dimethoxy-6-acetyl-2,2-dimethylchromene	5	Leaves and flowers	Isolation/CC	[42]
6-[Acetyl]-5,8-dimethoxy-2,2-dimethylchromene	6	N.A.	Acetylation of 2	[42]
2-Isopropenyl 6-methoxy-2,3-dihydrobenzofuran	7	Leaves and flowers	Isolation/CC	[42]
6-Methoxy 2-[1,2-dihydroxy-2-propyl] benzofuran	8a	Leaves and flowers	Isolation/CC	[42]
6-Methoxy-2-[1-acetoxy-2-hydroxy-2-propyl] benzofuran	8b	N.A.	Acetylation of 8a	[42]
7,8-Dimethoxy-2,2-dimethylchromene	9	N.A.	Hydrogenation of 1	[42]
7,8-Mimethoxy-[2,2-dimethyloxacetalhyde] benzaldehyde	10	N.A.	Ozonolysis of 1	[42]
Encecalinol	11	Aerial	HPLC Isolation/CC	[14] [16]
Encecalin	12	Aerial	HPLC Isolation/CC	[17] [16]
5-Acetyl-3*β*-angeloyloxy-2*β*-(1-hydroxyisopropyl)-2,3-dihydrobenzofuran	13	Aerial	Isolation/CC	[17]
5-Acetyl-3*β*-angeloyloxy-2*β*-(1-hydroxyisopropyl)-6-methoxy-2,3-dihydrobenzofurane	14	Aerial	Isolation/CC	[16]
*O*-Methylencecalinol	15	Aerial	Isolation/CC	[16]
Sonorol	16	Leaves	Isolation/CC	[16]
Encecanescin	17	Leaves	Isolation/CC	[20,21,43]
6-Acetyl-7-hydroxy-8-methoxy-2,2-dimethylchromene	18	Leaves	Isolation/CC	[44]
6-Acetyl-7-acetoxy-8-methoxy-2,2-dimethylchromene	19	N.A.	Acetylation of 18	[44]
6-Acetyl-7-hydroxy-8-methoxy-3-iodine-2,2-dimethyl-dimethylchromene	20	N.A.	Iodination of 18	[44]
3,5-Diprenyl-4-hydroxyacetophenone	21	Leaves	Isolation/CC	[21]
Espeleton	22	Aerial	Isolation/CC	[16]
(+)-*β*-eudesmol	23	Aerial	Isolation/CC	[16]
4,15-Epoxy-(+)-*β*-eudesmol	24	N.A.	Epoxidation of 23	[16]
4-Acetoxy-(+)-*β*-eudesmol	25	N.A.	Oxidation of 23	[16]
Taraxerol	26	Aerial	Isolation/CC	[16]
*β*-sitosterol	27	Aerial	Isolation/CC	[16]
Stigmasterol	28	Aerial	Isolation/CC	[16]
Chlorogenic acid	29	Leaves	HPLC	[26]
7-*O*-(*β*-D-glucopyranosyl)-gossypetin	30	Leaves	HPLC	[44]
7-*O*-(*β*-D-glucopyranosyl)-galactin	31	Leaves	HPLC	[44]
Nerol	32	Leaves	HydD/GC-MS	[28]
Bornyl acetate	33	Leaves	HydD/GC-MS	[28]
Thymol	34	Leaves	HydD/GC-MS	[28]
*β*-Elemene	35	Leaves	HydD/GC-MS	[28]
α-Humulene	36	Leaves	HydD/GC-MS	[28]
Germacrene-D	37	Leaves	HydD/GC-MS	[28]
Thymylisobutyrate	38	Leaves	HydD/GC-MS	[28]
Bicyclogermacrene	39	Leaves	HydD/GC-MS	[28]
8,9-Epoxithymylisobutyrate	40	Leaves	HydD/GC-MS	[28]
Eupatoriochromene	41	Leaves	HydD/GC-MS	[28]
α-Cadinol	42	Leaves	HydD/GC-MS	[28]
Androencecalinol	43	Leaves	HydD/GC-MS	[28]
Encecalol	44	Leaves	HydD/GC-MS	[28]
Ripariochromene-A	45	Leaves	HydD/GC-MS	[28]
Germacrene-A	46	Leaves	HydD/GC-MS	[28]
Cadinene	47	Leaves	HydD/GC-MS	[28]

N.A.: Not applicable; CC: Column chromatography; HPLC: High-performance liquid chromatography; HydD/GC-MS: Hydrodistillation/gas chromatography-mass spectrometry.

## Data Availability

Data sharing is not applicable to this article.

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
