# Peer review of "Phytochemical, Pharmacological, and Biotechnological Study of Ageratina pichinchensis: A Native Species of Mexico"

_plants, 2021, doi:10.3390/plants10102225_

Round 1

Reviewer 1 Report

2. Research Methodology
- one inclusion criteria - the authors of the papers that have been included in your analysis have been specified if the vegetal samples were analysed in order to establish the botanical identity

The chemical composition and and biological properties are very well presented.

The authors have some information about the ratio between the quantity of the most important compounds (with positive biological properties) ant the quantity of vegetal products for the cell culture method. 

Figure 5 is from scientific literature or is a personal study?

Reviewer 2 Report

The review manuscript presents a plant species of great interest, and the review is complete and very well written, presenting topics of clear chemical-biological importance.

Reviewer 3 Report

The manuscript “Phytochemical, Pharmacological and Biotechnological Study of Ageratina pichinchensis: a native species of Mexico” fits the journal’s scope. Although the review’s summary is well design, some sections are too briefly presented, and others are in too much detail (in these cases, the text is describing the methodology). Another major concern regarding this manuscript is the non-existence of a critical approach throughout the manuscript. The authors simply present some information here and there, and in many cases, the information is not even in the proper section. The whole manuscript needs extensive corrections regarding the English and the punctuation. Given the mention concerns (also please see the comments below), the manuscript should be corrected before another evaluation.   

Other comments:

Lines 37-38, 53-55, 316-317, 574-576 rephrase.

Line 43 – please correct the phrase, and be more specific. Many members of the Asteraceae family have flowers grouped in chapters.

66-67 – the authors should be very careful when stating this kind of affirmation, especially when the first inclusion criteria is that the article to be published in English.

Lines 126-127, 196, 229-230, 272-273, 300, 304 – please correct the errors.

Lines 229-261- the introduction for the Phenolic compounds subsection is too long. Please make it shorter.

Although the Phenolic compounds seem to be representative, and the authors state that are a main class of constituents for this species, this subsection should be re-organized. In the present form, there are few studies presented, some of them with too few details, whereas for other the method of extraction is described step by step. Also, a table with the phenolic compounds should be added.

Subsection 7.3 – here are discussed too few references. For example, why the data from reference 45 aren’t presented here? Please rewrite table 2 according to the structure adopted in table1.

Section 7: All subsections should be consistent. Thus, the authors should use other Ageratina species for exemplification in all subsections, or to refer only to A. pichinchensis.

329-332 please indicate only the abbreviations.

When applicable (e.g. subsection 8.1), the subsections which described the biological activities should be further organized in vitro, non-clinical and clinical tests.

411 please explain “validated”.

Section 8.2 – Here are presented few studies regarding the healing, antiulcer and potentially anticancer activity – Please re-organize the whole sub-section. Are necessary so many details in some cases (e.g. references 25, 27, 64)?

488-507 – the introduction of this section is too long.

Section 9 – this section presents only two references and the authors give too many details on the methodology used by others.

572-574 – How does this review demonstrate the biological properties???

Round 2

Reviewer 3 Report

The authors answered and clarified the majority of raised issues, and provided justification when it wasn’t possible to make modifications. The manuscript has been improved significantly, and now it is suitable for publication in Plants, after the following minor corrections:

Please redraw the structures in figure 7.

Lines 433-439 – these paragraphs are redundant and should be deleted or shortened.

Author Response

Responses to Reviewer 3 Comments

 Point 1. The authors answered and clarified the majority of raised issues and provided justification when it wasn’t possible to make modifications. The manuscript has been improved significantly, and now it is suitable for publication in Plants, after the following minor corrections:

Please redraw the structures in figure 7.

Response 1: This was done. The structures in Figure 7 were redrawn, Pag. 14. In addition, all the figures were checked and improved, and now have better resolution.

Point 2. Lines 433-439 – these paragraphs are redundant and should be deleted or shortened.

Response 2. This was done. The paragraph was revised, rewritten, and shortened. Currently lines 433-437.